# MicroRNA-Mediated Suppression of Glial Cell Line-Derived Neurotrophic Factor Expression Is Modulated by a Schizophrenia-Associated Non-Coding Polymorphism

**DOI:** 10.3390/ijms25084477

**Published:** 2024-04-19

**Authors:** Gergely Keszler, Bálint Vékony, Zsuzsanna Elek, Zsófia Nemoda, Nóra Angyal, Zsófia Bánlaki, Réka Kovács-Nagy, Zsolt Rónai, János M. Réthelyi

**Affiliations:** 1Department of Molecular Biology, Institute of Biochemistry and Molecular Biology, Semmelweis University, 1094 Budapest, Hungary; elek.zsuzsanna@semmelweis.hu (Z.E.); nemoda.zsofia@med.semmelweis-univ.hu (Z.N.); angyal.nora@med.semmelweis-univ.hu (N.A.); banlaki.zsofia@med.semmelweis-univ.hu (Z.B.); kovacs-nagy.reka@semmelweis.hu (R.K.-N.); ronai.zsolt@semmelweis.hu (Z.R.); 2Doctoral School, Semmelweis University, 1085 Budapest, Hungary; vekony.balint@stud.semmelweis.hu; 3Department of Psychiatry and Psychotherapy, Semmelweis University, 1083 Budapest, Hungary; rethelyi.janos@semmelweis.hu

**Keywords:** schizophrenia, glial cell line-derived neurotrophic factor (GDNF), 3′-untranslated region (3′-UTR), miRNA, genetic association analysis, single nucleotide polymorphism

## Abstract

Plasma levels of glial cell line-derived neurotrophic factor (GDNF), a pivotal regulator of differentiation and survival of dopaminergic neurons, are reportedly decreased in schizophrenia. To explore the involvement of GDNF in the pathogenesis of the disease, a case–control association analysis was performed between five non-coding single nucleotide polymorphisms (SNP) across the GDNF gene and schizophrenia. Of them, the ‘G’ allele of the rs11111 SNP located in the 3′ untranslated region (3′-UTR) of the gene was found to associate with schizophrenia. In silico analysis revealed that the rs11111 ‘G’ allele might create binding sites for three microRNA (miRNA) species. To explore the significance of this polymorphism, transient co-transfection assays were performed in human embryonic kidney 293T (HEK293T) cells with a luciferase reporter construct harboring either the ‘A’ or ‘G’ allele of the 3′-UTR of GDNF in combination with the hsa-miR-1185-1-3p pre-miRNA. It was demonstrated that in the presence of the rs11111 ‘G’ (but not the ‘A’) allele, hsa-miR-1185-2-3p repressed luciferase activity in a dose-dependent manner. Deletion of the miRNA binding site or its substitution with the complementary sequence abrogated the modulatory effect. Our results imply that the rs11111 ‘G’ allele occurring more frequently in patients with schizophrenia might downregulate GDNF expression in a miRNA-dependent fashion.

## 1. Introduction

Schizophrenia is a chronic neurodevelopmental disorder that starts in late adolescence or early adulthood and affects approximately 0.7% of the population worldwide [1]. Despite intensive research, its complex pathophysiological background is still far from being completely understood. Apart from the well-known deleterious role of perinatal noxae including fetal hypoxia, infection, undernourishment, smoking and drug abuse of the mother [2], the unusually high heritability of the disease (60–80%) implies a strong genetic determination that motivated numerous genetic studies. Among hundreds of candidate genes identified by recent genome wide association studies and fine-mapping of associated regions [3], some neurotrophic factors including brain-derived (BDNF) and glial cell-derived neurotrophic factors (GDNF) have received special attention due to their prominent role in the regulation of the development, differentiation, and survival of cortical neurons in general and dopaminergic neurons in particular [4,5]. Both the neurodevelopmental and dopamine theories of schizophrenia make neurotrophins potential susceptibility/candidate genes. Furthermore, there is compelling evidence for their therapeutic role in the treatment of psychiatric disorders including schizophrenia, addiction, and autism spectrum disorders [6].

GDNF belongs to the GDNF family of ligands (GFL), which are neurotrophic factors that signal by assembling ternary complexes with GFLα co-receptors and either members of the RET (rearranged during transfection) receptor tyrosine kinases or the neuronal cell adhesion molecule NCAM1. GDNF signaling prevents apoptosis and promotes the survival of dopaminergic and motor neurons by stimulating the ras/MAP kinase (mitogen activated protein kinase) and PI3K/Akt (phosphatidylinositol 3-kinase/protein kinase B) signaling pathways [7]. Apart from its essential functions in the central nervous system, special regulatory roles have been assigned to GDNF in spermatogenesis [8], kidney development [9] and liver fibrosis [10].

The revised dopamine hypothesis of schizophrenia postulates that cerebral dopamine imbalance due to suboptimal prefrontal dopamine signaling and excessive dopamine release from mesolimbic dopaminergic neurons in the ventral striatum elicits negative and positive symptoms in schizophrenia, respectively [11]. The first hints implicating potential correlations between GDNF and schizophrenia were based on the role GDNF is known to play in the development and survival of central dopaminergic neural circuitries at the cellular level [12]. In accordance with the seemingly logical presumption that neurodevelopmental disorders might be due to dysregulated GDNF signaling and hypofunction, initial studies shed light on decreased plasma GDNF levels in patients with schizophrenia either as a single marker [13], or in combination with other neurotrophic factors including BDNF, nerve growth factor (NGF) and Klotho [14]. Serum GDNF levels showed an inverse correlation with brief psychiatric rating scale (BPRS) scores in unmedicated patients with schizophrenia [4], while patients with deficit schizophrenia performing better on cognitive tests exhibited higher-than-average serum GDNF levels [5]. Importantly, antipsychotic treatment improved schizophrenia symptoms and enhanced GDNF levels as well [15,16]. However, the issue is not unequivocal as other studies found no correlation between plasma GDNF levels and schizophrenia [17,18], concluding that plasma GDNF concentration is an unreliable biomarker for schizophrenia.

Tissue GDNF levels in affected brain areas and in the cerebrospinal fluid (CSF) might seem physiologically more relevant markers than plasma levels. This assumption prompted some recent studies, but they also yielded contradictory results. Hidese et al. reported that GDNF levels in the CSF of patients with schizophrenia were lower than those in healthy individuals [19]. On the other hand, Mätlik et al. found elevated GDNF concentrations in CSF samples of first-episode psychosis patients, and, importantly, the expression of GDNF was also increased in the postmortem striatum of patients with schizophrenia [20].

Apart from plasma or brain GDNF levels per se, impaired GDNF signaling has also been implicated in the pathogenesis of schizophrenia. In support of this hypothesis, anti-NCAM1 autoantibodies disrupting the interaction of GDNF with NCAM1 could be detected in 5.4% of schizophrenia patients and injection of these antibodies precipitated schizophrenia-related symptoms in mice [21].

There have been numerous attempts to find genetic associations between the *GDNF* susceptibility locus and schizophrenia with moderate success. Lee et al. [22] deployed a single-strand conformational polymorphism (SSCP) analysis but did not find any single nucleotide variations associated with schizophrenia; however, they reported that an (AGG)_10_ short tandem repeat (microsatellite) polymorphism in the 3′ untranslated region of the *GDNF* gene was more common in patients with schizophrenia than in healthy controls. In agreement with that, high copy numbers (≥15) of this short tandem repeat were shown to confer protection from schizophrenia [23].

The first comprehensive genetic association analysis investigated a set of nine single nucleotide polymorphisms (SNPs) spanning the entire 40 kb of the *GDNF* locus in a large schizophrenia cohort [24]. Albeit two intronic SNPs were found to show nominal association with schizophrenia (*rs2973050*, *p* = 0.007; *rs2910702*, *p* = 0.039), the study could not recapitulate any correlation of the *AGG* length polymorphism with the disease. Importantly, Ma et al. [25] could not reproduce the above SNP associations in a 384-strong Chinese schizophrenia population. The latter study genotyped a total of seven tag SNPs in the gene, of which three were also analyzed previously, but could not confirm any statistically significant association with the disease. It is of note that genetic variants of the GFRA3 GDNF receptor have been shown to associate with schizophrenia as well [26].

It seems reasonable to assume that non-coding polymorphisms of the *GDNF* gene might modulate the risk of schizophrenia by influencing GDNF plasma levels, but the association studies cited above did not investigate this correlation. There is only one report of this kind of two novel and very rare 3′-UTR (3′ untranslated region) single nucleotide variants associated with elevated serum GDNF levels in patients suffering from bipolar disorder [27].

Recently, GDNF has been the focus of multiple genetic association analyses in our laboratory, revealing associations of certain SNPs in the gene with neuropsychiatric conditions including anxiety [28], depression [29], smoking [30], addiction [31], and gambling [32]. Based on this background, the aim of the present study was to seek genetic associations between non-coding SNPs in the *GDNF* gene and schizophrenia.

## 2. Results

### 2.1. Case–Control Association Analysis

As outlined in the Introduction, plasma GDNF levels seem to be reduced in patients with schizophrenia, but only scarce and ambiguous information is available to date on genetic factors modulating the expression of GDNF at the transcriptional and post-transcriptional levels. Therefore, it seemed particularly justified to explore whether some common (minor allele frequency [MAF] > 0.05) non-coding SNPs in the *GDNF* gene potentially influencing mRNA stability are associated with schizophrenia. Here, we took advantage of a well-characterized control group already used and genotyped in one of our previous *GDNF*-related genetic association studies [28]. Five non-coding tag SNPs (four of them located in intronic sequences and the fifth one in the 3′-UTR of the last exon; Table 1) were selected from the eight-strong SNP set genotyped earlier [28] with similar spacing along the *GDNF* gene to cover five of its six conserved haplotype blocks (Figure 1). These SNPs were genotyped in a 275-strong cohort of schizophrenia patients using quantitative polymerase chain reaction (qPCR)-based TaqMan assays (Table 1). Genotype and allele frequencies in the case and control cohorts as well as corresponding confidence values from χ2 statistical analyses are shown in Table 2. Genotype distributions of all polymorphisms were in Hardy–Weinberg equilibrium (HWE). Of the five SNPs investigated, only the *rs11111* in the 3′-UTR was found to significantly associate with schizophrenia both in the context of genotype (*p* = 0.006) and allele distribution (*p* = 0.001). The *p* value for genotype distribution remained significant upon FDR (false discovery rate) and Hochberg correction for multiple testing (the cutoff value in both tests was 0.006), while the allele distribution remained significant even in the more stringent Bonferroni and Holm corrections too (cutoff: 0.001). These statistically firm data led us to conclude that the minor (G) allele of this polymorphism occurred more frequently among schizophrenia patients than in healthy controls.

As schizophrenia often manifests itself with a heterogeneous spectrum of symptoms of varying severity, it seemed reasonable to analyze the correlation of alleles of the *rs11111* SNP with symptoms and their severity assessed by means of the Positive and Negative Symptom Scale (PANSS) in personal interviews led by professional psychiatrists. Domain-specific symptom severity was assessed using the five-factor model of PANSS (positive and negative symptoms, cognitive impairment, hostility, and depression) [34]. The ‘G’ allele of the rs11111 polymorphism showed a significant correlation with positive symptoms (*p* = 0.0320) and cognitive factors (*p* = 0.0405) (Table 3), of which only the former correlation passed the Hochberg correction for multiple testing but not the Bonferroni, Holm, and FDR correction analyses. These results further reinforced our notion that the minor allele of the *rs11111* SNP might be a risk factor in the development of schizophrenia through an elusive molecular mechanism that deserved further elucidation.

### 2.2. Functional Analysis

#### 2.2.1. In Silico Prediction of microRNA Binding

Localization of the *rs11111* SNP within the 3′-UTR of the gene raised the intriguing possibility that it might modulate RNA interference, a post-transcriptional gene regulatory mechanism based on microRNA binding to the target mRNA. To test this hypothesis, an in silico search was performed to explore whether allelic variants of the *rs11111* SNP might affect putative miRNA binding sites. According to the Polymorphism in microRNA Target Site (PolymiRTS) database, the presence of the *rs11111* ‘G’ allele creates an optimal, perfectly matching seed region for three different microRNAs: *hsa-let-7f-2-3p*, *hsa-miR-1185-1-3p*, and *hsa-miR-1185-2-3p*, which are all expressed in the gray matter of the brain [35]. Proposed hybridization patterns of these miRNAs to the 3′-UTR of the *GDNF* mRNA are illustrated in Figure 2. “R” denotes the position of the *rs11111* A/G SNP. The critically important seed region of miRNAs (nucleotides 2 to 8) is shown in bold and nucleotides complementary to the 3′-UTR sequence outside the seed region are underlined. As can be seen in the figure, the seed region of all three microRNAs is completely complementary with the 3′-UTR in the presence of the *rs11111* ‘G’ allele only. The *hsa-miR-1185-1-3p* and -*2-3p* ‘twin’ species differ in a non-complementary single pyrimidine nucleotide in position 20 solely (indicated in italics), which is located outside the seed region, resulting in a complementarity of 64% for both miRNAs, while that of the *hsa-let-7f-2-3p* is only 38%, making its interaction with the *GDNF* mRNA thermodynamically far less favorable. Moreover, according to the Human miRNA Tissue Atlas database, expression levels of *hsa-let-7f-2-3p* in the central nervous system are much lower than that of the two other microRNAs, rendering it a less likely candidate to regulate GDNF expression in vivo. As far as the biological relevance of the twin microRNAs is concerned, *hsa-miR-1185-1-3p* was one of the four microRNAs to be differentially expressed in induced pluripotent stem cell (iPSC)-astrocytes derived from schizophrenia patients and healthy controls [33], while no literary information is available linking the other two microRNAs to schizophrenia. Considering these arguments, we opted to test *hsa-miR-1185-1-3p* in transient reporter assays.

#### 2.2.2. Transient Transfection-Based Reporter Assays

In light of the above in silico results, we assumed that binding of *hsa-miR-1185-1-3p* to mRNA harboring the ‘G’ (but not the ‘A’) allele might downregulate GDNF levels. To experimentally validate this hypothesis, luciferase reporter-based transient transfection assays were performed.

To generate the reporter vector, an 800 bp long segment of the 3′-UTR of the *GDNF* gene harboring the ‘G’ allele of the *rs11111* SNP was PCR-amplified from genomic DNA and subcloned into the pMIR-REPORT^TM^ reporter vector. The ‘A’ allelic variant was generated by site-directed mutagenesis. Two further constructs, a deletion mutant lacking the entire seed region of the miRNA target sequence and an inverse mutant featuring the complementary (antisense) sequence of the seed region, were also created as putative negative controls of microRNA action.

Different amounts of the reporter plasmids were transfected into confluent cultures of HEK293T human embryonal kidney cells using the Lipofectamine 2000 reagent. HEK293T cells are easy to transfect and according to the Human miRNA Tissue Atlas, *hsa-miR-1185-1-3p* is not expressed in the kidney, enabling us to exclude any modulatory effect of endogenously produced miRNA. To normalize the transfection efficiency, constant amounts of a *β*-galactosidase expressing reporter plasmid were also co-transfected into the cells. The modulatory effect of *hsa-miR-1185-1-3p* was assessed by co-transfecting 0, 1, 5, or 25 pmol of its commercially available precursor. To exclude any dose-dependent non-specific effect of miRNA, *hsa-miR-1185-1-3p* was applied in combination with *hsa-miR-20b* in a way that the sum of the amount of both miRNa species was always 25 pmol in the transfections. According to the miRBase database (https://www.mirbase.org (accessed on 7 September 2021)), *hsa-miR-20b* shows no sequence similarity to *miR-1185* so any interference in binding to the GDNF mRNA could be ruled out. Luciferase and *β*-galactosidase levels were assayed by luminometry and colorimetry from freeze–thaw crude extracts, respectively. Normalized reporter activities are shown in arbitrary units, taking the luciferase levels of microRNA-free samples as 100% (Figure 3 and Figure 4). *p* values were calculated from relative activity differences with ANOVA (analysis of variance).

The transient transfection assays provided clear-cut evidence in support of the allele-specific inhibitory effect of hsa-miR-1185-1-3p as increasing amounts (1, 5 and 25 pmol) of the microRNA suppressed normalized reporter activities in a dose-dependent manner with statistically significant (*p* = 0.0075) downregulation in the presence of 25 pmol microRNA (Figure 3A). On the other hand, luciferase levels of the other construct harboring the ‘A’ allele were not influenced by the microRNA (Figure 3B). Moreover, as expected, neither the deletion nor the inverse mutants exerted any regulatory effects on the reporter activities (Figure 4A,B).

## 3. Discussion

MicroRNAs have long been known to fine-tune gene expression at the post-transcriptional level, giving rise to complex gene regulatory networks [36]. Recently, various novel mechanisms have emerged that further modulate the interaction between microRNAs and their cognate target sequences, including polymorphisms in miRNA target sequences in 3′ gene regulatory regions [37], polymorphisms affecting the primary sequence of microRNAs [37,38], sequestration of microRNAs by non-coding circular RNAs [39], and endogenous sponging of microRNAs by the presence of alternative binding sites in target genes [40]. Although microRNAs are studied mostly in human malignancies [37], their regulatory roles are increasingly recognized in the pathogenesis of neuropsychiatric disorders too [41]. In this study, a schizophrenia-associated SNP affecting a microRNA binding site in the 3′-UTR of the *GDNF* gene has been functionally characterized.

The seminal role of GDNF in the maintenance and survival of dopaminergic neurons has impelled several studies aiming to treat Parkinson’s disease with this neurotrophic factor [42]. However, the importance of GDNF cannot be underestimated in psychiatric disorders either, as a significant association was uncovered with certain sequence variants and/or plasma levels of GDNF in anxiety and depression [30], attention deficit hyperactivity disorder [43], Tourette syndrome [44], and substance dependence [31]. As dysfunctional dopamine signaling is a hallmark of schizophrenia, several studies have addressed the involvement of dopaminergic neurons in general and GDNF in particular in schizophrenia. Despite a number of case–control and genome-wide association studies (GWAS), only two single nucleotide variants and a trinucleotide repeat polymorphism in the *GDNF* gene were shown to nominally associate with schizophrenia [22,23,24]. However, the association of these SNVs could not be reaffirmed by another study [25], and no functional studies were run either.

To our best knowledge, the present study is the first that (1) sheds light on any genetic association of the *rs11111* SNP localized within the 3′ untranslated region of the *GDNF* gene and (2) addresses the functional role of any genetic polymorphism in the *GDNF* gene. Having shown a statistically firm genetic association between the *rs11111* SNP and schizophrenia, we provided evidence that the ‘G’ allele occurring more frequently in patients with schizophrenia might promote the binding of an inhibitory microRNA to *GDNF* mRNA, resulting in downregulation of GDNF levels. Interestingly, the *rs11111* locus was also nominally associated with positive symptoms generally ascribed to altered dopamine signaling in the striatal–temporolimbic system, hinting at a potential link between *GDNF* sequence variants, GDNF expression, and dopaminergic neurotransmission in schizophrenia.

The putative regulatory role of the unusually long (almost 3 kbp) 3′-UTR in the *GDNF* gene has long been studied. Initial transient reporter assays revealed that a 200 bp segment located 1000 bp downstream of the stop codon represses gene expression [45]. The inhibitory effect of the 3′-UTR on GDNF expression was confirmed in vivo by generation of transgenic mice with conditional knockout [46] and CRISPR/Cas9-mediated deletion [47] of a major part of the 3′-UTR, which resulted in increased but spatiotemporally conserved GDNF expression. Providing insight into the inhibitory mechanism, the former paper was the first report on miRNA-dependent regulation of GDNF expression, presenting unambiguous evidence that *miR-9*, *-96*, *-133* and *-146a* downregulated reporter activities in transiently transfected HEK293 and U87 cells [46]. Later on, it turned out that a further cohort of miRNAs can bind to the 3′-UTR and fine-tune gene expression, including *miR-17-5p* [48], *miR-33* [49,50], *miR-204* [51], and *miR-451* [52]. Interestingly, GDNF signaling has been shown to modulate the expression levels of various miRNA precursors in a MAPK-dependent fashion [53], raising the intriguing possibility of the existence of mutual GDNF–miRNA regulatory circuitries. These findings highlight the complexity of the regulation of GDNF expression at the post-transcriptional level, which is further enriched by our results presented here.

Our luciferase reporter assays functionally confirmed that the presence of the minor ‘G’ allele creates a binding site that perfectly matches the seed region of *hsa-miR-1185-1-3p* as verified by downregulation of luciferase levels in the case of the reporter construct harboring the ‘G’ but not the ‘A’ allele. The fact that this microRNA is widely expressed in the hippocampus and various cortical areas [35] implies that *hsa-miR-1185-1-3p* might regulate GDNF expression in the brain, and individuals possessing the ‘G’ risk allele express lower amounts of GDNF levels, rendering them more vulnerable to the disease as early as during embryonic development, a critical period according to the neurodevelopmental hypothesis of schizophrenia. Although dozens of miRNAs have been proposed to be differentially expressed in schizophrenia [54], this is the first study implicating hsa-*miR-1185-1-3p* in the disease with a functional role mediated by a binding site polymorphism.

Recently, *miR-1185-1* species have been implicated in the pathogenesis of human malignancies including urinary bladder tumors [55] and colorectal cancer [56,57], as well as in obesity-related inflammation [58] and Alzheimer’s disease [59]. Importantly, the last three studies cited here not only found an association of *miR-1185-1* levels with the disease but also made fruitful attempts to identify target genes at the molecular level using both in silico tools and cell-based reporter assays.

In their exquisite paper, Wang et al. [57] provided convincing evidence that SIRT1 (Sirtuin 1) downregulates *miR-1185-1* levels in colorectal cancer cells by deacetylating its promoter, which results in enhanced expression of CD24, a target gene promoting colon cancer cell stemness. They performed dual-luciferase reporter assays in HT29 colorectal adenocarcinoma cells with a reporter construct harboring the 3′-UTR of the *CD24* gene. Upon co-transfection of cells with *miR-1185-1*, a 26% reduction was seen in reporter activities in the presence of the canonical seed region 5′-CTGTATA-3′ in the reporter that is commensurate with what we could see in the case of our ‘G’ reporter construct (Figure 3A, 30% suppression with 25 pmol miRNA). In line with our observations, this inhibitory effect was abrogated when a mutant reporter with a scrambled seed sequence was deployed.

Garcia-Lacarte and co-workers [58] identified *miR-1185-1* as a predictor of response to dietary restriction and a bona fide post-transcriptional regulator of glycogen synthase kinase 3β (GSK3B), a pro-inflammatory signal transducer. In 3′-UTR dual-luciferase assays, administration of *miR-1185-1* to HEK293T cells repressed reporter activities over 50%, a profound decrease that might be ascribed to the fact that the *GSK3B* 3′-UTR contains two cognate binding sites for the miRNA species, and both were subcloned into the reporter construct. Unexpectedly, however, there was no significant inhibition of reporter expression with *miR-548q* despite the presence of a putative full-match seed binding site in the 3′-UTR. This finding shows how necessary it is to confirm the functional relevance of seemingly sound miRNA–mRNA associations predicted solely by bioinformatic tools.

Of the *miR-1185-1* related functional investigations published so far, the study of Delay et al. [59] seems to bear the most relevance to our results. Using in silico algorithms, they predicted that the *rs9909* C/G SNP polymorphism in the 3′-UTR of the nucleoporin 160 (*NUP160*) gene, which was shown to associate with Alzheimer’s disease, affects the seed region of the *miR-1185-1-3p* binding site, with the ‘G’ allele creating a perfect binding site while the ‘C’ variant partly abrogates that. The setting is therefore highly similar to that of the *rs11111* A/G SNP in the *GDNF* gene: in both cases, the 3rd nucleotide of the seed region is polymorphic, with the ‘G’ allele being the token of full complementarity. Importantly, co-transfection of HEK293T cells with 3′-UTR reporter constructs and *miR-1185-1-3p* resulted in significantly reduced luciferase levels in the presence of the ‘G’ but not in that of the ‘C’ allele. It should be noted that *miR-1185-1-3p* slightly enhanced reporter levels in the case of the ‘C’ allele, an interesting but unexpected observation made by us when performing the experiment with the non-complementary ‘A’ allele (Figure 3B). These data are in good agreement with our findings, confirming the critical regulatory role of the ‘G’ allele, embedded in the third position of the same seed region in the context of two functionally unrelated genes that are assumed to be involved in the pathogenesis of neuropsychiatric diseases. Although the occurrence of functional SNPs in miRNA binding sites is not unprecedented at all [60], the fact that the seed region of the same miRNA localized in two different 3′ untranslated domains is modulated by the same allele of two different SNPs can, in our opinion, be considered a rarity.

## 4. Materials and Methods

### 4.1. Participants

Our case–control study involved 275 in- and outpatients with schizophrenia (mean age: 37.7 ± 11.6 years, 46.6% males) and 708 age- and gender-matched healthy controls (mean age: 21.33 ± 3.39 years, 46.3% males). Both cohorts were of Caucasian origin. The control group consisting of university students and other volunteers has already been characterized in detail in a former study of ours [28]. Detailed genotype data of the control cohort were therefore available for all SNPs investigated in the present study. Patients with schizophrenia were recruited at the Department of Psychiatry and Psychotherapy of the Semmelweis University; the diagnosis was made according to DSM IV (Diagnostic and Statistical Manual of Mental Disorders IV) diagnostic criteria and the MINI 5.0 Neuropsychiatric Interview [61]. Exclusion criteria included neurological disorders, head trauma, substance use, and mental retardation. Symptom severity and complexity were assessed by means of the Positive and Negative Symptom Scale (PANSS) instrument [34].

### 4.2. Principles of SNP Selection

SNPs with a minor allele frequency (MAF) greater than 0.05 were selected from the Single Nucleotide Polymorphism database of NCBI (dbSNP). The pairwise tagging method using an r^2^ threshold of 0.8 by Haploview was used to determine tagging SNPs based on HapMap data to obtain proper coverage of the GDNF gene. SNPs with a reference from previous association studies in relation to neuropsychiatric disorders were preferred.

### 4.3. DNA Sampling and Purification

Genomic DNA was prepared from patients’ peripheral blood samples as follows. 450 μL of blood sample were mixed with 450 µL of proteinase K buffer (0.1 M NaCl, 0.01 M Tris-HCl pH = 8, 0.5% SDS, 0.2 mg/mL proteinase K) and incubated at 56 °C overnight. Proteins were then precipitated using saturated NaCl and removed by centrifugation. DNA was isolated from the supernatant using the standard ethanol/isopropanol precipitation protocol. Precipitated DNA was redissolved in 100 μL of Tris–EDTA solution (0.005 M Tris–HCl, pH = 8 and 0.5 mM EDTA) overnight at 4 °C. Genomic DNA from control individuals was prepared from buccal swabs as described in [28]. DNA concentrations were quantitated with a Nanodrop1000 spectrophotometer. Samples with a greater than 15 ng/μL DNA concentration were diluted fivefold. DNA stocks obtained this way were diluted a further 15-fold for qPCR-based genotyping.

### 4.4. SNP Genotyping

Predesigned SNP genotyping assays containing 2 primers to amplify the adjacent region of the polymorphisms and 2 allele-specific TaqMan probes labeled with FAM and VIC fluorescent dyes, respectively, were obtained from Thermo Fisher (Waltham, MA, USA). Assay IDs are provided in Table 1. Quantitative PCR reaction mixtures contained 3 μL of TaqMan ProAmp master mix (AmpliTaq Gold^®^ DNA polymerase, dNTPs, ROX dye, MgCl_2_ and buffer), 0.15 μL TaqMan SNP kit, 1.85 µL PCR-grade water and 1 μL genomic DNA in a final volume of 6 µL. Polymerase chain reactions were initiated with a 95 °C–10 min step to activate the hot-start AmpliTaq Gold^®^ DNA polymerase, followed by 40 thermocycles of 95 °C–15 s denaturation and 60 °C–1 min combined annealing and extension, respectively. Reporter signal was detected during this latter step and an endpoint allelic discrimination analysis was also performed after PCR amplification to classify the samples into 3 clusters according to their genotype. Thermocycling was performed with a Thermofisher 7300 LightCycler in 96-well plates.

### 4.5. In Silico Tools

Genomic sequences were downloaded from the National Center for Biotechnology Information (NCBI) Nucleotide database (https://www.ncbi.nlm.nih.gov/nucleotide/ (accessed on 7 January 2021.)). Primers amplifying a segment of the GDNF 3′-UTR that spans the rs11111 SNP were designed with the NCBI Primer Blast tool (https://www.ncbi.nlm.nih.gov/tools/primer-blast (accessed on 8 January 2021.)). MicroRNAs displaying allele-specific binding to this sequence were identified with the help of the PolymiRTS database (https://compbio.uthsc.edu/miRSNP (accessed on 7 January 2021)). Tissue-specific expression patterns of miRNA species were obtained from the Human miRNA Tissue Atlas (https://ccb-web.cs.uni-saarland.de/tissueatlas2 (accessed on 21 February 2021)) [35].

### 4.6. Construction of Reporter Vectors

An 800 bp long segment of the 3′-UTR of the GDNF gene harboring the rs11111 polymorphic locus was PCR-amplified from the genomic DNA sample of an rs11111 ’GG’ individual using sequence-specific primers (forward: 5′-TGTCGTGAGCTCCACTTCCTGTTGT; reverse: 5′-CCCGCCAAGCTTCTTCCTCCTGCT; PCR steps: 94 °C–10 min enzyme activation, 40 cycles comprising 95 °C–15 s denaturation, 60 °C–1 min annealing and 72 °C–1 min extension). The amplicon was cloned in between the HindIII and SacI restriction sites of the pMIR-REPORT^TM^ vector (restriction cleavage sites in the primers are underlined). FastDigest restriction enzymes were from New England Biolabs, Ipswitch, MA, USA (8 × 10^−8^ unit/reaction, incubation: 37 °C for 10 min). Ligation was performed with T4 DNA ligase (New England Biolabs, 16 °C for 12 h). An aliquot of the ligase reaction was transformed into XL10 Gold^®^ *E. coli* ultracompetent cells (Agilent™, Santa Clara, CA, USA) according to the manufacturer’s instructions and selected in the presence of 100 mg/mL carbenicillin. Plasmids were isolated from 24 h colonies with the PureYield™ Mini plasmid kit (Promega, Madison, WI, USA) and sequenced by Microsynth Inc., Balgach, Switzerland. Sequencing confirmed that the obtained clone contained the rs11111 ‘G’ allele. The A-containing allelic variant was generated with the Agilent™ Quikchange Lightning site-directed mutagenesis kit in order to create constructs completely isogenic apart from the rs11111 polymorphic locus. Primers for mutagenesis were designed with an online tool of the manufacturer (https://www.agilent.com/store/primerDesignProgram (accessed on 26 February 2021)). The following primer pairs were used to generate the A-containing allele: forward: 5′-TCAGGTGTTTGGGTATACAGGAGCAGCAGCTGTTG; reverse: 5′-CAACAGCTGCTGCTCCTGTATACCCAAACACCTGA; primers for the seed complementary mutant were as follows: forward: 5′-CAGGGAGCTGTCAGGTGTTTGGGATATATCGAGCAGCAGCTGTTGACCCCCGG; reverse: 5′-CCGGGGGTCAACAGCTGCTGCTCGATATATCCCAAACACCTGACAGCTCCCTG. Primers to generate mutants with deleted seed regions were designed according to the protocol described by Liu and Naismith [62] (forward: 5′-CTCCCCAAACACCTGACAGCTCCCTGGGGAGCAGATG; reverse: 5′-GTCAGGTGTTTGGGGAGCAGCAGCTGTTGACCCCC). All constructs were verified by sequencing prior to transfection (Microsynth Inc., Balgach, Switzerland).

### 4.7. Cell cultures and Transfection

The human HEK293T cell line was maintained in Dulbecco’s modified essential medium (DMEM, Invitrogen—Gibco, Waltham, MA, USA) complemented with 10% fetal blood serum (Lonza) and antibiotics (100 µg/mL streptomycin and carbenicillin, respectively) at 37 °C in a humidified atmosphere containing 5% CO_2_. Log-phase cells were seeded in 24-well plates 24 h prior to transfection. Cells were co-transfected with 0.005 or 0.025 µg luciferase reporter plasmid, 0.600 µg *β*-galactosidase reporter plasmid as an internal control, and 0, 1, 5 or 25 pmol of hsa-miR-1185-1-3p pre-miRNA (Merck), co-incubated for 20 min in 7.5 µL Lipofectamine 2000 reagent (Invitrogen) and diluted to a final volume of 125 µL with OptiMEM™ medium (Invitrogen—Gibco, Waltham, MA, USA). For proper dosage compensation in each assay, hsa-miR-1185-1-3p was complemented with hsa-miR-20b to a total amount of 25 pmol miRNA. Each transfection was performed in triplicate and repeated three times in independent assays. After transfection, cells were incubated for 48 h, then washed and collected in phosphate-buffered saline (PBS) solution.

### 4.8. In Vitro Reporter Assays

Cells were extracted by three consecutive freeze–thaw cycles (liquid N_2_ and 37 °C water bath). Luciferase activities were assayed by adding 60 μL Luciferin reagent (0.16 mg/mL Luciferin K, 20 nM Tricine, 2.6 nM MgSO_4_, 0.1 nM Na_2_EDTA, 33.3 nM dithiothreitol (DTT), 0.27 nM Li_3_CoA and 0.53 nM Na_2_ATP) to 12 μL of crude extracts. Luminescence was measured using a Varioskan multi-well plate reader (Thermo Fisher Scientific Inc., Waltham, MA, USA). *β*-galactosidase activity was assayed by colorimetry using ortho-nitrophenyl-β-D-galactopyranoside (ONPG) as a chromogenic substrate. Luciferase levels were normalized using *β*-galactosidase activity as an internal control.

### 4.9. Statistical Analysis

The Hardy–Weinberg equilibrium (HWE) for genotype distributions was assessed with the χ2-test. Association analyses were carried out by comparing genotype frequencies of each polymorphism in all patient cohorts using SPSS v17.0 and HaploView v4.2 [63]. To rule out false positive results, an online adjustment tool for multiple correction testing was used (https://multipletesting.com (accessed on 7 October 2020)), enabling us to perform simultaneous Bonferroni, Holm, Hochberg and false discovery rate (FDR) analyses [64]. Linkage disequilibrium and haplotype analyses were performed with the HaploView v4.2 software (https://mybiosoftware.com/haploview-4-2-analysis-visualization-ld-haplotype-maps.html (accessed on 07 October 2020)). Genotype data of a healthy, Caucasian population for linkage disequilibrium testing were obtained from the 1000 Genomes Browser (https://www.internationalgenome.org/1000-genomes-browsers/index.html (accessed on 7 October 2020)).

Association analysis between PANSS subscale factors was carried out with the General Linear Model (GLM) algorithm of the Statistical Analysis System’s SAS/STAT software version 14.2. ANOVA of the luciferase assays was performed with the GraphPad InStat software (version 3.05).

## 5. Conclusions

In this case–control study we found that the rs11111 SNP in the 3′-UTR of the GDNF gene is statistically associated with schizophrenia susceptibility in general and with its positive and cognitive symptoms in particular. In silico tools revealed that the G allele of this polymorphism might create a binding site for three microRNAs widely expressed in the brain. In in vitro transient transfection assays, hsa-miR-1185-1-3p downregulated reporter activities in the presence of the G allele. Limitations of this study include the relatively small sample size of the schizophrenia cohort and the use of a non-neuronal cell line in the reporter assays. Further studies are needed to better characterize the importance of this modulatory effect in a more physiological context, including recapitulation of the reporter assays in neural cell lines and measuring endogenous GDNF transcript levels in pre-miRNA treated cells.

## Figures and Tables

**Figure 1 ijms-25-04477-f001:**
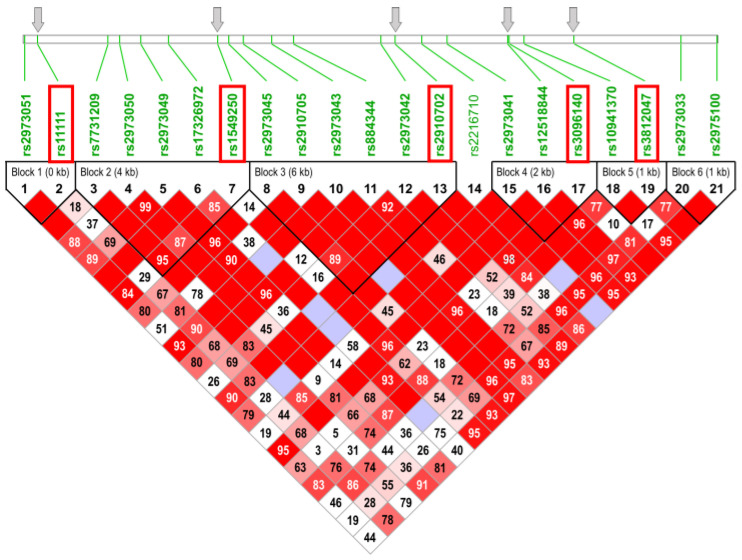
Representation of haplotype blocks with over 90% linkage disequilibrium (LD) in the GDNF gene. Tag SNPs genotyped in the study are in red boxes and their relative positions are shown by thick arrows. The image was generated with Haploview from data from the HapMap database.

**Figure 2 ijms-25-04477-f002:**
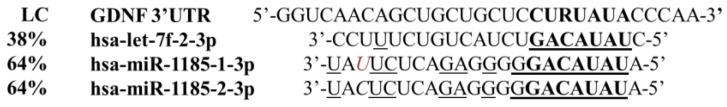
In silico sequence alignment of candidate microRNAs to the 3′-UTR of the *GDNF* mRNA. The seed region is printed in bold, and ’R’ marks the position of the *rs11111* A/G SNP within. miRNA nucleotides complementary to the target sequence are underlined. Nucleotide 20 in *hsa-miR-1185-1-3p* and *-2-3p* (U versus C), marking the only sequence difference between them, is in italics. Levels of complementarity (’LC’) were calculated by dividing the number of complementary nucleotides by the total length of miRNAs.

**Figure 3 ijms-25-04477-f003:**
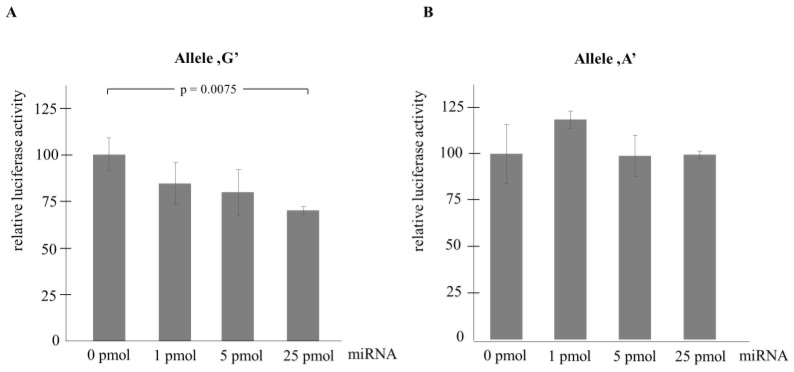
Relative luciferase activities obtained with ‘G’ (**A**) and ‘A’ allele (**B**) containing reporter vectors. Transient transfection assays were run in triplicate. Cells were co-transfected with 0.025 g of luciferase reporter vector, 0.6 µg of *β*-galactosidase vector, and increasing amounts of pre-hsa-miR-1185-1-3p as indicated. Luciferase levels were normalized to *β*-galactosidase activity and are shown in relative units as compared to those measured in cells untreated by pre-microRNA (left bar). The only significant difference between relative luciferase levels is indicated.

**Figure 4 ijms-25-04477-f004:**
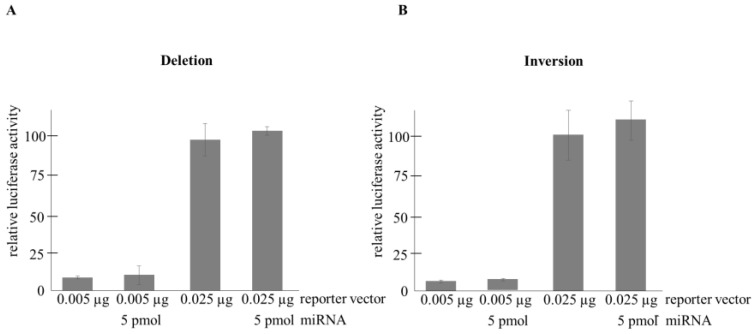
Relative luciferase activities obtained with deleted (**A**) or inverse (**B**) seed region containing reporter constructs. Transient transfection assays were run in triplicate. Cells were co-transfected with 0.005 or 0.025 µg of luciferase reporter vector, 0.6 µg of *β*-galactosidase vector, and 0 or 5 pmol of *pre-hsa-miR-1185-1-3p* as indicated. Luciferase levels were normalized to *β*-galactosidase activity and shown in relative units as compared to those measured in cells transfected with 0.025 µg reporter but no miRNA (bar 3).

**Table 1 ijms-25-04477-t001:** Characterization of single nucleotide polymorphisms (SNPs) genotyped in this study. SNPs are listed in the order of their genomic location taken from the Genome Reference Consortium human 38 (GRCh38) database. Note that the GDNF gene is in the antisense orientation on the chromosome. Literary minor allele frequencies are from the gnomAD browser database (https://gnomad.broadinstitute.org (accessed on 13 April 2020.)) and correspond to data obtained in a European population.

SNP	Allele	Genomic Location (GRCh38)	Intragenic Location	MAF	TaqMan ID	References
rs11111	A/G	5:37814000	3′ UTR	0.16 (G)	C___8813050_1_	[28,30]
rs1549250	G/T	5:37821119	intronic	0.42 (G)	C__11553504_10	[28,30,31]
rs2910702	A/G	5:37828202	intronic	0.25 (G)	C__15948353_10	[24,25,28,30]
rs3096140	C/T	5:37832731	intronic	0.30 (C)	C___1395038_20	[28,30,33]
rs3812047	A/G	5:37835296	intronic	0.13 (A)	C__27492935_10	[28,29,30]

**Table 2 ijms-25-04477-t002:** Genotype and allele distributions for all polymorphisms investigated in the case and control cohorts. HWE, *p* values calculated for Hardy–Weinberg equilibrium.

SNP	Genotype	Genotype Frequency (%)		HWE		*p* Value	Allele Frequency (%)		*p* Value
		Control	Patient	Control	Patient	Genotype	Control	Patient	Allele
rs11111	AA	75.9	66.2	0.132	0.44	**0.006**	86.7	80.9	
	AG	21.7	29.5						**0.001**
	GG	2.4	4.3				13.3	19.1	
rs1549250	TT	33.5	33.3	0.806	0.799	0.988	57.7	57.3	
	TG	48.4	48.1						0.957
	GG	18.1	18.6				42.3	42.7	
rs2910702	AA	54.8	59.9	0.667	0.557	0.360	74.3	77.0	
	AG	38.9	34.1						0.157
	GG	6.3	6.0				25.7	23.0	
rs3096140	CC	48.0	51.1	0.700	0.132	0.329	69.5	70.0	
	CT	43.0	37.8						0.424
	TT	9.0	11.1				30.5	30.0	
rs3812047	AA	76.6	74.3	0.781	0.785	0.755	87.4	86.4	
	AG	21.8	24.0						0.494
	GG	1.7	1.7				12.6	13.6	

**Table 3 ijms-25-04477-t003:** Association of rs11111 genotypes and alleles with schizophrenia factors. Significant *p* values are shown in bold.

SNP	Positive Factor	Negative Factor	Hostility Factor	Cognitive Factor	Depression Factor
rs11111 AG	**0.0320**	0.2669	0.4243	**0.0405**	0.2466

## Data Availability

The data presented in this study are available upon request from the corresponding author.

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
