# Peer review of "MicroRNA-Mediated Suppression of Glial Cell Line-Derived Neurotrophic Factor Expression Is Modulated by a Schizophrenia-Associated Non-Coding Polymorphism"

_ijms, 2024, doi:10.3390/ijms25084477_

Round 1
Reviewer 1 Report
Comments and Suggestions for Authors
The article by Gergely Keszler and colleagues investigates the potential role of microRNA (miRNA)-mediated regulation of glial cell line-derived neurotrophic factor (GDNF) expression in the context of schizophrenia susceptibility. The research focuses on the ‘G’ allele of the rs11111 SNP in the 3' untranslated region (3'-UTR) of the GDNF gene, potentially leading to the downregulation of GDNF in patients with schizophrenia. This study further suggests that while the exact mechanisms are complex, one potential pathway involves variations within the GDNF gene, mainly single nucleotide polymorphisms (SNPs) located in non-coding regions.
The main question explored by this study is to investigate the connection between specific variations in the GDNF gene, particularly the rs11111 SNP, and its potential influence on GDNF expression through microRNA-mediated mechanisms, ultimately exploring a potential link to schizophrenia susceptibility.
The study focuses on the interaction between the rs11111 SNP, miRNA binding, and GDNF expression, which is potentially relevant, as it delves into a specific mechanism potentially affecting GDNF levels.
The authors have used SNP genotyping, In silico tools, construction of reporter vectors, cell transfection, and in vitro reporter assays for the study's methodology.
This study needs to incorporate the following things to strengthen the article.
-
Authors should include more recent literature related to this study and discuss this in detail in the “Discussion” section.
-
The authors need to conclude the study's findings, limitations, and future prospects by adding a “Conclusion” section to the manuscript.
-
The authors should mention the full forms for the first time, e.g., for “FDR.”
-
In Figure 1, the text is not clearly visible. Authors should provide clear and readable images.
-
The authors need to be consistent with the nomenclature of the cell line for “HEK293T” or “HEK293”.
-
The language is clear and professional throughout, but a few minor grammatical errors and awkward phrasings could be smoothed out. e.g., “(Fug, 3, panel B)” on line 251.
-
In Figure 4 description, the authors have mentioned, “Cells were co-transfected with 0.005 or 0.025 µg of luciferase reporter vector, 0.6 µg of β-galactosidase vector and 0 or 5 pmol of pre-hsa-miR-11851-3p as indicated” but in the actual figure, I didn't see “0 pmol” mentioned. This needs to be clarified by the authors.
Author Response
On behalf of all co-authors, I would like to express our sincere thanks for your thorough review of our manuscript and the constructive criticism intended to improve its quality. Please find below our detailed responses to all your remarks and comments.
1. Authors should include more recent literature related to this study and discuss this in detail in the „Discussion” section.
A new paragraph has been included in the Discussion section, introducing the role of 3’-UTR and miRNA polymorphisms in the post-transcriptional regulation of the expression of neurotrophic factors in general and GDNF in particular, in light of most recent scientific findings.
2. The authors need to conclude the study’s findings, limitations, and future prospects by adding a „Conclusion” section to the manuscript.
A „Conclusion” section specifying the above has been added to the manuscript.
3. The authors should mention the full forms for the first time, e.g., for „FDR”.
We maximally agree with your remark; full forms are to be provided for better understanding of the terms and text. Abbreviations have been resolved for the first time not only for FDR but also for 13 additional terms such as RET, PolymiRTS, GWAS, qPCR, PI3K/Akt, ANOVA, SIRT1, GSK3B, NUP160, DSM IV, NCBI, DTT and ONPG.
4. In Figure 1, the text is not clearly visible. Authors should provide clear and readable images.
Figure 1 has been redrawn with far better resolution and clearly readable characters. SNPs genotyped in the study have been highlighted both by arrows and red boxes.
5. The authors need to be consistent with the nomenclature of the cell line for „HEK293T” or „HEK293”.
Transient transfection experiments were performed using HEK293T cells. The nomenclature is now used consistently and unequivocally throughout the manuscript.
6. The language is clear and professional throughout, but a few minor grammatical errors and awkward phrasings could be smoothed out. e.g., „(Fug. 3, panel B)” on line 251.
Thank you for noticing this typing error. This and some further typing errors have been rectified and the text has also been streamlined from a grammatical point of view.
7. In Figure 4 description, the authors have mentioned, „Cells were co-transfected with 0.005 or 0.025 mg of luciferase reporter vector, 0.6 mg of β-galactosidase vector and 0 or 5 pmol of pre-hsa-miR-1185-1-3p as indicated” but in the actual figure, I didn’t see „0 pmol” mentioned. This needs to be clarified by the authors.
We apologise for omitting the ’0 pmol’ inscription. It is shown in the revised version of Fig. 4 under bars 1 and 3.
We hope that amendments outlined above meet your expectations and make the manuscript suitable for publication.
Sincerely yours,
Gergely Keszler
corresponding author
Reviewer 2 Report
Comments and Suggestions for Authors
In this study, Kezler et al. examined the role of the GDNF-associated SNP rs11111 in schizophrenia and whether it may have a functional role in schizophrenia biology. First, they showed that of the several GDNF SNPs tested, the G allele of rs11111 in the 3’UTR is significantly associated with schizophrenia and two of five symptoms tested. The authors performed an in silico analysis, which showed that three miRNAs may bind to the 3’-UTR region. Using an in vitro reporter assay, the authors showed that hsa-miR-1185-1-3p interacted with the G allele in a dose-dependent manner. Overall, the study is interesting in linking a SNP to schizophrenia, albeit in a relatively small sample size, and showing the functional significance of the SNP in its potential regulation by miRNAs.
Major comment:
What is the expression of these miRNAs in the brain? In particular, the authors tested hsa-miR-1185-1-3p. Is this miRNA or other ones identified in silico expressed in the brain? Does reference 52 (Alzheimer’s Disease) show expression of hsa-miR-1185-1-3p in brain tissues? The authors should show either from literature or from brain-derived cell lines that these miRNAs are specifically expressed in the brain. This is important because if these miRNAs are not expressed in the brain, it undermines their narrative that the G allele is functionally associated with schizophrenia.
Minor comments:
Please clarify whether rs11111 has been implicated by the schizophrenia GWAS.
Line 165 and Table 3: The hostility factor (or cognitive factor) has been mislabeled.
Line 251: “Fig. 3” instead of “Fug, 3”
For Figure 4, also label 0 pmol miRNA and add more detail to the title. For instance, instead of just inversion, mention “inversion of 3’UTR seed sequence.”
Include the relatively small size of the cohort as a limitation.
Author Response
On behalf of all co-authors, I would like to express our sincere thanks for your thorough review of our manuscript and the constructive criticism intended to improve its quality. Please find below our detailed responses to all your remarks and comments.
Major comment
We maximally agree that it is very important indeed to know whether hsa-miR-1185-1-3p is expressed in the brain as it could confer biological (in vivo) relevance on our results. Though we have not checked it in our study due to unavailability of human brain tissue or brain-derived cell lines, but there is compelling literary evidence in favor of explicit cerebral expression of has-miR-1185-1-3p as follows:
-
Authors of reference [52] did not check cerebral levels of miR-1185-1-3p either but cited two references in their paper which showed altered expression levels of this miRNA in the brain of Alzhemeir’s patients:
Lau et al., Alteration of the microRNA network during the progression of Alzheimer's disease. EMBO Mol Med 2013 Oct;5(10):1613-34. doi: 10.1002/emmm.201201974
Hébert et al., A study of small RNAs from cerebral neocortex of pathology-verified Alzheimer's disease, dementia with lewy bodies, hippocampal sclerosis, frontotemporal lobar dementia, and non-demented human controls. J Alzheimers Dis 2013;35(2):335-48. doi: 10.3233/JAD-122350.
Both papers provide clear-cut evidence that hsa-miR-1185-1-3p is expressed in the healthy brain while its expression is profoundly reduced in Alzheimer’s disease.
2. On the other hand, one can find high-resolution data on expression levels of hsa-miR-1185-1-3p in various brain areas in the miRNATissueAtlas2 database [Keller et al., miRNATissueAtlas2: an update to the human miRNA tissue atlas Nucleic Acids Res 2022 Jan 7;50(D1):D211-D221. doi: 10.1093/nar/gkab808] at https://ccb-web.cs.uni-saarland.de/tissueatlas2/patterns/hsa/mirna/hsa-miR-1185-1-3p/.
Importantly, relative expression of miR-1185-1-3p is the second highest in the brain amon gall tissues investigated, and it is particularly enriched in the cortical gray matter of all lobes, making it a biologically relevant candidate to regulate GDNF brain levels in vivo.
As far as the other two miRNAs are concerned, miR-1185-2-3p is also expressed in the brain and significantly enriched in the hippocampus and pituitary gland. Hsa-let-7f-2-3p is also expressed in all areas of the brain but less abundantly than miR-1185-1-3p and -2-3p.
This information has been summarized briefly in the manuscript.
Minor comments
1. Please clarify whether rs11111 has been implicated by the schizophrenia GWAS.
The rs11111 SNP has not been implicated in schizophrenia by genome-wide association studies. Two other SNPs occurring in the GDNF gene, rs2973050 and rs2910702 have been shown to nominally associate with the disease (reference [24] in the manuscript) but these associations were not recapitulated by another study (reference [25] in the manuscript). To make this statement unambiguous in the text, these SNPs have been written in the corresponding sentence in the Discussion of the manuscript.
2. Line 165 and Table 3: The hostility factor (or cognitive factor) has been mislabeled.
Thank you for raising our attention to the mislabeling. rs11111 is associated with positive and cognitive symptoms of schizophrenia. Both the text and headings of Table 2 have been corrected accordingly.
3. Line 251: „Fig. 3” instead of „Fug. 3”
Unfortunately, a small number of typing errors including the one you have pointed to remained in the manuscript. They have been carefully corrected.
4. For Figure 4, also label 0 pmol miRNA and add more detail to the title. For instance, instead of just inversion, mention „inversion of 3’UTR seed sequence.”
Omission of hsa-miR-1185-1-3p has been clearly indicated in Fig. 4 as ’0 pmol’ under bars 1 and 3 in both panels. Headings of Fig. 3 and 4 have been modified as Seed sequence with allele ’G’, Seed sequence with allele ’A’, Deletion of 3’-UTR seed sequence and Inversion of 3’-UTR seed sequence, respectively, according to your suggestions.
5. Include the relatively small size of the cohort as a limitation.
It has also been advised by Reviewer #1. Accordingly, we have added a ’Conclusion’ section to the manuscript that includes this statement too.
We hope that amendments outlined above correspond to your expectations and make the manuscript suitable for publication.
Sincerely yours,
Gergely Keszler
corresponding author